# Impact of supervisory behavior on sustainable employee performance: Mediation of conflict management strategies using PLS-SEM

**Jiang Min**[1], **Shuja Iqbal**[1]*, **Muhammad Aamir Shafique Khan**[1]*, **Shamim Akhtar**[1], **Farooq Anwar**[2], **Sikandar Ali Qalati**[1]

**1** School of Management, Jiangsu University, Zhenjiang, P.R. China, **2** Lahore Business School, University of Lahore, Lahore, Pakistan

* drkhanju@hotmail.com (KMAS); shujaiqbal88@hotmail.com (SI)

**Data Availability Statement:** All relevant data are within the manuscript and its Supporting Information files.

## Abstract

This study investigates the relationship between supervisory behavior, conflict management strategies, and sustainable employee performance and inquires the mediating effect of conflict management strategies. Data were collected from the SMEs of the manufacturing industry of Pakistan. The significance of the model was assessed using the PLS-SEM (structural equation modeling). The findings of the study revealed a positive and significant relationship between supervisory behavior and sustainable employee behavior. Similarly, conflict management strategies had a positive effect on the relationship between supervisory behavior and sustainable employee behavior. This study adds in the current literature of supervisory behavior as a critical predictor of sustainable employee performance in two ways. Firstly, this study validates Conflict management strategies as an influential mediator between the relationship of supervisory behavior and sustainable employee performance. Secondly, this study provides substantial practical implications for managers at SMEs to enhance sustainable employee performance through supervisory behavior, stimulated by conflict management strategies. This study is based on cross-sectional data; more longitudinal studies can further strengthen the generalizability of relationships between the constructs. The study adds in the current literature of PLS-SEM as an assessment model for direct and mediation relationships.

## 1. Introduction

SMEs in Pakistan are under extensive research in recent years because of their valuable contribution to the country's economic growth and provision of employment opportunities [1]. SMEs contribute more than 99 percent of the business in the country leading to a significant share in manufacturing exports (25 percent). Overall, SMEs contribute 30 percent of all exports, contribute to value addition by 28 percent, and create employment opportunities. Moreover, forty percent of the GDP of the country comes from SMEs [2].

The phenomena of aggressive competition among the firms have escalated the necessity to achieve sustainable employee performance (SEP), through conflict management strategies

**Funding:** The authors received no specific funding for this work.

**Competing interests:** The authors have declared that no competing interests exist.

(CMS), steered by the supervisory behavior (SB). Albeit many factors contribute to determining SEP, the relationship among the supervisor and sub-ordinates plays a critical role. Supervisors have a significant impact on the organizations they lead [3]. Employee performance mends through SB, especially by coaching and team management activities [4]. Indeed, SB needs to be considered in small and medium enterprises (SMEs) in Pakistan; where, leadership positively predicts employee performance [1]. Also, SB crucially impacts employees' task achievement and retention [5]. The supervisor's behavioral elements are critical for SEP; namely, perceptual discrepancy, supportive behavior, value congruity, trustworthiness, and similar personalities have a link to employee performance and job satisfaction. In Pakistan's context, similar characters, supportive behavior, and perceptual discrepancies were found positively influencing job satisfaction [6]. Persistently, SEP plays a critical role in SMEs concerning limited human capital and high-performance requirements. Accordingly, SB and CMS are imperative constituents of SEP. Elements of SB may be useful techniques for managers at SMEs to achieve SEP in their organizations.

Based on the background of employees' job performance and organizational performance [1], this study adds to the SEP literature. This study focuses on finding the impacts of SB on SMEs' SEP. Various preceding studies focused on the direct effects of SB on outcomes, for instance, employee performance [4], organizational commitment [5], job satisfaction [6], salesforce participation [7], and path-goal relationships [8]. Likewise, previous research has focused on direct effects of CMS on outcomes, such as intragroup conflict perceptions [9], organizational performance, corporate governance, and employee performance [10], and efficacy and employee performance [11]. However, the mediating role of CMS between the relationship of SB and SEP was less prominently examined in past studies. This study addresses this gap in the literature and proposes a model, including SB and CMS, explaining SEP. Although the significant direct relationships in SB, CMS, and employee performance were found, more empirical studies on the mediating role of CMS are essential. Moreover, similar research relating to SMEs in the country are rare. Therefore, this empirical study would expand the understanding of SEP with elements of SB and CMS.

Furthermore, this study uses the partial least squares structural equation modeling (PLS-SEM) for data analysis. SEM is a statistical method that presents the intricate relationships powerfully and conveniently [12–14] developed PLS path modeling, which explained the variance of endogenous latent variables optimized by approximating partial model relationships. PLS path model uses latent variables scores estimated as exact linear groupings of their related manifest variables and treats them as error-free alternates for manifest variables [15].

This study has the following contributions. We used an inclusive approach to explore the multifaceted mediation role of CMS on the relationship between SB and SEP. Past studies on CMS mostly examined its' direct effects on constructs such as efficacy and performance [11], innovation performance [16], interpersonal rewards and performance [17], job satisfaction, cognition of action-taking, and sleep disorder [18]. Moreover, previous studies have found the positive relationship of SB and CMS [19–22], employee performance and SEP [23, 7, 3, 24]. However, this study uniquely examines the mediation role of CMS between the relationship of SB and SEP. Finally, this study enriches the literature on SB, CMS, and SEP and provides substantial practical implications for managers at SMEs to enhance SEP.

## 2. Literature review and hypotheses development

### 2.1 Literature review

SMEs stand significantly in the economic well-being of both developing and developed countries. Notably, in developing countries, SMEs play a vital role in attaining "sustainable

development goals." SMEs create job opportunities, nurture inventions, reduce income dissimilarities, and stimulates sustainable industrialization [1]. In Pakistan, SMEs contribute 90% of the business (most of them operate in the easy-going, undocumented sector). SMEs have significant value in addition to the economic growth of the country by job creation. On the other hand, SMEs are considered as the poverty elevation mechanism, in terms of providing jobs to lower-income groups of the country. SMEs contribute around 40% to GDP and provides 80% of the industrial workforce, and contribute one fourth in export earning of the sector [25–27]. Concerning the fundamental role of SMEs in the economy of Pakistan, this study focused on SMEs to examine the impacts of SB on SEP.

SB in SMEs is critical for employee performance [1]. According to the ability, motivation, and opportunities (AMO) theory [28], employee performance is highly linked with these three factors. Previous research has examined the AMO model as a vital tool to create effective performance management systems [29]. This study examined CMS as an *ability* of the human resource department (supervisors in particular) to stimulate employee performance (vice versa). Moreover, this study focused on the *motivation* of subordinates to delicately handle SB and *opportunities* linked with CMS to achieve SEP.

**2.1.1 Sustainable employee performance.** Over the last decades, a phenomenal increase in the importance of sustainable organizations was examined. The conception of sustainability emerged from "ecology," denoting the capability of organizations and procedures to cultivate, raise, care, and to sustain [30]. Past research has presented the idea of sustainable work performance and the impact of management and organizational practices on SEP [31]. Likewise, job performance has been a significant area in human resource management practices. Job performance is the level of an employee's contribution to the effectiveness of a firm concerning the specific performance benchmarks associated with his/her job [32]. Several factors significantly contribute to job performance, to name a few, job-related factors, environmental and firm related factors, and employee-related factors [33–36]. Related to the idea of job performance, past studies have particularly examined SEP. Sustainable performance refers to the exclusive efforts of employees for personal and organizational sustainable growth. Sustainable individual task performance and relational development were considered significant measures of SEP [3].

**2.1.2 Supervisory behavior.** Research into SB has a rich history in various fields. The behavior of the supervisor includes moral and professional support, building sound workplace, assistance towards improving subordinates' performance. SB includes several elements of a leader's behavior towards subordinates, such as perceptual discrepancy, supportive behavior, value congruity, trustworthiness, and similar personalities affecting employee performance [6]. The skills associated with the SB significantly impact employee's performance. For instance, person-oriented and task-oriented supervisory skills are substantial towards job satisfaction and employee retention. Multiple studies have examined the impact of SB on outcomes such as employees' moods and psychological well-being [37, 23] supervisor support on employee retention [34]; SB (transformational leadership) on employee retention [38] similarly, studies have found lower levels of job satisfaction in the result of abusive supervision. Past research has thoroughly examined core concepts related to SEP affected by SB (leadership). For instance, aversive leadership negatively linked to job satisfaction [39], a substantial relationship in job satisfaction and leadership style [40], ethical leadership and other related concepts [41–43]. It was known to be a significant influence on the work behavior and morale of employees [44]. This study proposes that SB affects SEP substantially.

**2.1.3 Conflict management strategies.** Conflicts are inevitable in any workplace. The performance of employees could be damaged, collapsed, and incompetent by conflicts [45]. Past research examined both negative and positive impacts of conflicts, positive in terms of group thinking and status quo [46]. Such outcomes divided conflicts into destructive and non-

destructive (constructive) types [47]. Conflict management plays a vital role in conflict resolution, as [47] defined that conflict management is a "behavior oriented toward intensification, reduction, and resolution of tension." Scholars suggested that a certain level of conflict is vital to encourage creativity, avoid stagnation, and enhance employee performance. There are different types of conflicts based on the nature of conflict itself, such as "manifest, perceived, latent, line and staff, organized and unorganized conflicts" [19]. Also, studies show that CMS plays a significant role in the relationship between employees' teams and organizational commitment [48].

Organizations strive to achieve constructive conflicts rather than destructive conflicts to succeed. CMS plays an essential role in managing conflicts and avoid turning them into destructive ones. Past research has examined several types of CMSs such as [49] stated "resignation, isolation, withdrawal and cover-up" in avoiding procedures and "fighting, compromise, arbitration and negotiation" in functional strategies. Similarly, detailed plans include "competing, collaborating, compromising, avoiding and accommodating" [50]. This study examined the mediation of CMS between the relationships of SB and SEP.

**2.1.4 Partial least square structural equation model (PLS-SEM).** This study uses SEM as a standard reporting method to allow replicability and establish rigor. SEM is a second-generation multivariate data analysis approach that tests theoretically maintained linear and additive causal models [51, 52]. Researchers can examine relationships among the constructs using SEM. SEM is ideal for analyzing direct and indirect effects because it can measure hard-to-measure and unobservable latent variables. SEM consists of two models; the inner model examines the relationships between dependent and independent latent constructs, and the outer model examines the relationships between latent constructs and their observed indicators. PLS-SEM focuses on the analysis of variance, which could be carried out using SmartPLS. PLS is a soft modeling method for SEM, which has no assumptions about data distribution [53]. Therefore, PLSE-SEM becomes a suitable substitute to CB-SEM when the sample size is small, predictive precision is vital, little theory available on application, and correct model provisions cannot be ensured [54, 55].

Past research has used PLS-SEM in multiple disciplines such as strategic management and marketing [56], operation and international management [57, 58], accounting [59], tourism [60], family business [61], organization and group research [62]. Moreover, this study used PLS-SEM instead of covariance-based SEM to predict the dependent affected by latent variables. Furthermore, an increasing trend was seen in published articles using PLS-SEM [63]. The rationale for the choice of PLS-SEM was as follows. First, in CB-SEM, the scores of latent variables are indeterminant, which makes it unsuitable to use in predictive studies. In comparison, PLS-SEM produces "a single determinant score for each SEM composite for each observation" [64]. Secondly, in CB-SEM, $R^2$ relates to the proportion of common variance explained. In contrast, in PLS-SEM $R^2$ relates to the total variance explained [65].

## 2.2 Hypotheses development

**2.2.1 The relationship of SB and SEP.** Past studies have investigated several elements of SB concerning employee performance. Such as adaptive SB affects salesforce's performance. The psychological well-being of employees enhances through supervisory support ensuing in healthier workplaces [23]. A literature review by [7] examined multiple dimensions of SB related to employee performance, such as "closeness of supervision, frequency of communication, consideration, initiation of structure, feedback behavior, reward, punishments, and coaching behavior." [3] examined the positive effects of SB (transformational leadership style) on SEP in the construction industry of China. Positive and significant impacts of SB

(coaching) were found on job satisfaction and performance of warehouse employees in past studies [24]. A study by [66] found significant impacts of SB, such as reward and penalty power on subordinate's quality of work. Supportive supervisory reward behavior and advancement behavior of supervision were found significantly consistent by [67].

Previous studies indicated significant impacts of SB's dimensions, such as similar personalities, supportive behavior, and perceptual discrepancies on job satisfaction in SMEs of Pakistan [6]. The similarity in the personalities of supervisors and subordinates leads to higher levels of employee performance. A similar personality dimension substantially develops rewarding relationships among supervisors and employees. However, studies also have suggested that for instructions-based supervision, only similar personalities are not sufficient [68]. Supportive behavior of supervisors (by precise job tasks and dividing work equally) reduces job stress and leads to job satisfaction [69], which leads to SEP. Similarly, perceptual discrepancies between supervisors and subordinates create higher levels of job dissatisfaction and quitting intentions [70], resulting in the bad performance of employees. Therefore, this study proposes the impact of SB on SEP.

*Hypothesis 1: SB positively and significantly impacts SEP.*

**2.2.2 The relationship of SB and CMS.**   Supervisors demonstrate skills to adopt and implement CMS to manage conflicts in organizations. Such strategies may be a part of the SB or established and implemented by the top management of the organizations. Initially, scholars have suggested that supervisors manage conflict through avoidance, accommodation, competition, collaboration, and compromise [19, 20]. Past research has shown a strong relationship in the SB (how they treat their subordinates) and employee's performance in a conflict. In such situations, employees think their supervisors have less expectations from them, resulting in avoidance. Scholars suggested that supervisors use multiple types of CMS to handle conflicts concerning different types of employees. Using only one strategy to deal with various conflicts may not help in achieving the desired outcomes. Supervisors represent their organizations; thus, their style of coping with conflict does not only affect supervisor-subordinate relationships but employees' loyalty and trust as well. Supervisors can access the type of conflict and choose an appropriate strategy of conflict resolution. Unresolved conflicts can result in severe consequences [71, 21] discussed in their literature review study that leaders who involve in investigating conflict often choose assertive approaches to handle the conflict. In contrast, the leaders involved in the creation of conflict are suppler in managing the conflict as they do not care about the outcomes. Evidence from past research has shown a positive and significant link between SB and their choice of CMS, for instance, "problem-solving, compromising, dominating, and avoiding" strategies [22] therefore, this study proposes that:

*Hypothesis 2: SB has a positive and significant link with CMS.*

**2.2.3 The relationship of CMS and SEP.**   Past research has examined the positive impacts of CMS on SEP. [72] identified a positive relation of CMS, such as "joint consultation, mediation, collective bargaining, conciliation, arbitration," and employee performance in public sector organizations in Nigeria. Research showed the impacts of CMS on various outcomes, including efficacy and performance in teams, measuring through cooperative, competitive, and perception of high conflict efficacy approaches [11]. CMS "integrating, accommodating, compromising, forcing, avoiding were examined for innovation performance in previous studies [16]. Other studies have also examined effects of CMS on outcomes such as performance and interpersonal rewards and system rewards [17], innovation performance [16], job satisfaction, sleep disorder, and cognition of action taking [18]. This study focuses on the CMS based on [19]; Competing involves the desire of satisfaction of one party's conflict, irrespective of its impacts on the other parties. The collaborating strategy consists of satisfying the interest of all the parties concerned with a specific conflict. Avoiding technique suggests managers and

employees to prevent and escape from a situation of conflict or stress. Accommodating allows one party to place other party's interests above its' interest to resolve the matter of disagreement. The compromising strategy suggests both parties give up something to settle each other's conflicts. This study proposes the relationship as follows:

*Hypothesis 3: CMS positively and significantly affects SEP.*

**2.2.4 Mediation of CMS.**   Conflicts are inescapable in any workplace. Supervisors and employees can have conflicts on many issues, yet these issues are meant to be resolved in the best possible way to achieve organizational goals effectively. Supervisors adopt several CMSs by using their skills and behavior to implement those strategies in the best way to enhance employees' performance. Studies examined the effects of SB, such as communication, to adopt an appropriate CMS to enhance employees' performance [73]. Past studies examined relationships in CMS and job performance of employees [74]. Section 2.2.1 discussed the impacts of SB on employees' performance. Similarly, section 2.2.2 discussed definite links between SB and CMS, and section 2.2.3 addressed the effects of CMS on SEP. considering the positive relationships in SB, CMS, and SEP, this study uniquely proposes the mediating effect of CMS between the relationship of SB and SEP.

*Hypothesis 4: CMS positively and significantly mediates the relationship between SB and SEP.*

[19] examined that supervisors' behavior of using competing, compromising, accommodating, collaborating strategies was preferred by employees in different situations to resolve conflicts and enhance employee performance. Supervisors' collaborative conflict management behavior plays a significant decisive role in organizations and vice versa [75, 76]. Studies also suggest that collaborating and accommodating strategy enhances performance rewards [17, 77]. Supervisors need to choose specific strategies such as competing, compromising, collaborating, accommodating, and avoiding resolving conflicts which enhances employee performance [78]. Conflict management by strategies of supervisor enhances employees' performance [79]. Thus, we propose that CMSs positively mediates between the relationship of SB and SEP (all the hypothesis are presented in Annexure 1).

*Hypothesis 5a: Competing CMS positively and significantly mediates the relationship between SB and SEP.*

*Hypothesis 5b: Collaborating CMS positively and significantly mediates the relationship between SB and SEP.*

*Hypothesis 5c: Compromising CMS positively and significantly mediates the relationship between SB and SEP.*

*Hypothesis 5d: Avoiding CMS positively and significantly mediates the relationship between SB and SEP.*

*Hypothesis 5e: Accommodating CMS positively and significantly mediates the relationship between SB and SEP.*

Based on the above discussion, this study proposes the research model in Fig 1.

## 3. Research methodology

A quantitative method was adopted for this study based on a survey questionnaire. The study was conducted in the Punjab province, with the highest proportion of SMEs at 65.4 percent [27]. There are forty-eight thousand industrial units in the province of Punjab [80]. Past research presents a threshold that the sample size of 30 to 500 is sufficient for analysis [81]. Thus, this study collected data from 150 employees of SMEs using the simple random sampling technique consistent with studies of Arshad, Rasli, [82–84]. The questionnaires were sent online to the respondents. However, 122 functional questionnaires were received (81%

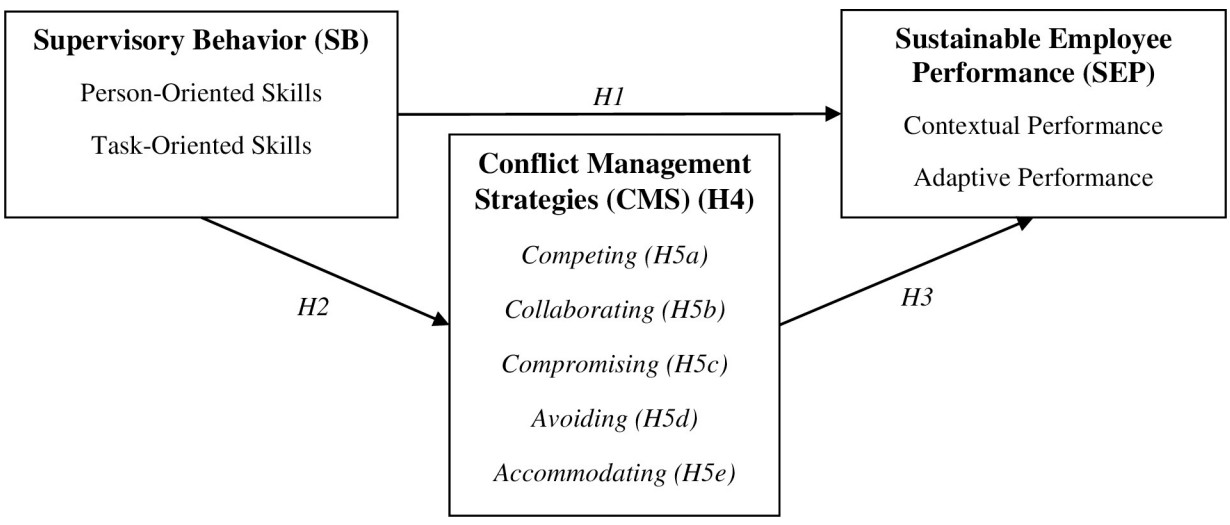

**Fig 1. Proposed research model and hypotheses.**

response rate). The study focused on employees as the subject to examine the effects of supervisory behavior on sustainable employee performance. Table 1 exhibits the demographic information of the survey respondents, including gender, age, and industry, and work experience.

## 3.1 Data analysis

This study used partial least squares (PLS) modeling to analyze the conceptual model. We used PLS path modeling because it has received a vast application in management and related fields [56, 85, 86]. This study aimed to predict the dependant variable; hence PLS path modeling was considered a suitable investigative method [15]. Scholars suggest PLS as the "most fully developed and general system" [87] about the "variance-based structured equation modeling" method [88]. Therefore, the data were further analyzed using Smart-PLS 3 to examine the proposed relationships.

**Table 1. Demographic information.**

| Controls | | Variance |
|---|---|---|
| Gender | Male | 78 (64%) |
| | Female | 44 (36%) |
| | 20–30 years | 23 (18.9%) |
| | 31–40 years | 47 (38.5%) |
| | 41–50 years | 36 (29.5%) |
| | >50 years | 16 (13.1%) |
| Industry | Manufacturing | 96 (78.7%) |
| | High-tech | 7 (5.7%) |
| | Construction | 13 (10.7%) |
| | Services | 6 (4.9%) |
| Experience | 1–5 years | 29 (23.8%) |
| | 6–10 years | 41 (33.6%) |
| | 11–15 years | 34 (27.9) |
| | >15 years | 18 (14.8%) |

## 3.2 Measures

In the present study, three variables were to be measured, including SB, CMS, and SEP.

*Supervisory behavior*: to measure SB, this study adopted from [89], developed by [90]. Two dimensions, including person-oriented skills (5 items) and task-oriented skills (3 items), were adapted from the past study. Cronbach alpha for the scale was satisfactory (0.902).

*Conflict Management Strategies*: this study adopted by measurement scales from [19] to measure CMS. Five strategies, including competing, collaborating, compromising, avoiding, and accommodating, were chosen (3 items each). Cronbach alpha for the scale was satisfactory (0.957).

*Sustainable Employee Performance*: this study focused on two dimensions of performance, including contextual performance (4 items) and adaptive performance (3 items) adopted from the study of [91]. These two dimensions described the best possible link with the conflict situations and individual efforts of employees to improve their performance. The scale was added with the meaning of "sustainable," and the Cronbach alpha for the scale was satisfactory (0.931).

## 3.3 Ethical statement

This study involved human participants and was reviewed and approved by the ethics committee of Lahore Business School, University of Lahore, Pakistan.

## 4. Results

### 4.1 Reliability and validity

**4.1.1 Measurement model assessment.** As suggested by past studies for assessing the measurement model, individual item reliability, Cronbach alpha, convergent reliability were utilized [92].

*Individual Item reliability (Loadings)*: outer loadings of each item for all constructs are suggested to determine the individual item reliability [93, 94]. Past studies provided a threshold that individual items' reliability should be equal to or more than 0.70 [92]. In the present study, all of the individual item reliabilities are 0.715 or more, exhibited in Table 2. Thus, the study meets the criteria for individual item reliability.

*Composite reliability (CR)*: or internal consistency reliability has been threshold by researchers to be 0.7 or above [95, 15]. As Table 2 shows that composite reliability of each item for the present study ranges between 0.916–0.961, adequate internal consistency in all constructs was measured.

*Convergent validity (AVE)*: [96] recommended average variance extracted to measure the convergent validity. The threshold was recommended by [97] that *AVE* should be at least 0.50 or above to measure each construct's convergent validity. The *AVE* for all the constructs has achieved the minimum level of 0.50 [98, 63], resulting in sufficient convergent validity of constructs used in this study (ref Table 2).

*Cronbach alpha (CA)*: the present study achieved the rule of thumb for Cronbach alpha values of 0.70 to 0.90 [98], as depicted in Table 2.

*Discriminant Validity (DV)*: [96] criteria were followed to assess the convergent validity following the rule of thumb of *AVE* value, which should be 0.50 or higher. They have also presented a rule of thumb for discriminant validity that *AVE's* square roots should be higher than the correlation in latent variables. All the *AVE* values are above the threshold of 0.50 (ref Table 2), and the square roots of *AVE* are higher than the correlation among latent variables (ref Table 3). Findings show that there were positive relationships among the variables of this

**Table 2. Measurement model.**

| Construct | Item code | Loading | Outer Weights | P-value | CA | CR | AVE |
|---|---|---|---|---|---|---|---|
| **Supervisory Behavior (SB)** | | | | | 0.902 | 0.925 | 0.674 |
| | SB-POS1 | 0.715 | 0.173 | <0.000 | | | |
| | SB-POS2 | 0.816 | 0.202 | <0.000 | | | |
| | SB-POS3 | 0.878 | 0.218 | <0.000 | | | |
| | SB-TOS1 | 0.832 | 0.209 | <0.000 | | | |
| | SB-TOS2 | 0.836 | 0.192 | <0.000 | | | |
| | SB-TOS3 | 0.838 | 0.221 | <0.000 | | | |
| **Conflict Management Strategies (CMS)** | | | | | 0.957 | 0.961 | 0.625 |
| *Accommodating Strategy* | ACC1 | 0.873 | 0.367 | <0.000 | 0.864 | 0.917 | 0.787 |
| | ACC2 | 0.910 | 0.397 | <0.000 | | | |
| | ACC3 | 0.878 | 0.363 | <0.000 | | | |
| *Avoiding Strategy* | AVO1 | 0.914 | 0.369 | <0.000 | 0.890 | 0.932 | 0.820 |
| | AVO2 | 0.897 | 0.370 | <0.000 | | | |
| | AVO3 | 0.906 | 0.365 | <0.000 | | | |
| *Collaborating Strategy* | COL1 | 0.937 | 0.344 | <0.000 | 0.934 | 0.958 | 0.884 |
| | COL2 | 0.954 | 0.369 | <0.000 | | | |
| | COL3 | 0.928 | 0.351 | <0.000 | | | |
| *Compromising Strategy* | COM1 | 0.901 | 0.390 | <0.000 | 0.868 | 0.919 | 0.791 |
| | COM2 | 0.875 | 0.360 | <0.000 | | | |
| | COM3 | 0.892 | 0.374 | <0.000 | | | |
| *Competing Strategy* | CPG1 | 0.896 | 0.364 | <0.000 | 0.875 | 0.923 | 0.800 |
| | CPG2 | 0.886 | 0.366 | <0.000 | | | |
| | CPG3 | 0.900 | 0.389 | <0.000 | | | |
| **Sustainable Employee Performance (SEP)** | | | | | 0.931 | 0.945 | 0.709 |
| | SEP-A1 | 0.857 | 0.178 | <0.000 | | | |
| | SEP-A2 | 0.825 | 0.174 | <0.000 | | | |
| | SEP-A3 | 0.829 | 0.167 | <0.000 | | | |
| | SEP-C1 | 0.831 | 0.159 | <0.000 | | | |
| | SEP-C2 | 0.898 | 0.179 | <0.000 | | | |
| | SEP-C3 | 0.854 | 0.173 | <0.000 | | | |
| | SEP-C4 | 0.797 | 0.156 | <0.000 | | | |

CA (Cronbach Alpha), CR (Composite Reliability), AVE (Average Variance Extracted)

**Table 3. Discriminant validity (Latent variable correlation and square root of AVE).**

| | ACC | AVO | CMS | COL | COM | CPG | SB | SEP |
|---|---|---|---|---|---|---|---|---|
| ACC | **0.887** | | | | | | | |
| AVO | 0.614 | **0.906** | | | | | | |
| CMS | 0.831 | 0.838 | **0.905** | | | | | |
| COL | 0.673 | 0.699 | 0.903 | **0.940** | | | | |
| COM | 0.697 | 0.676 | 0.902 | 0.767 | **0.890** | | | |
| CPG | 0.679 | 0.686 | 0.901 | 0.768 | 0.800 | **0.894** | | |
| SB | 0.710 | 0.711 | 0.843 | 0.730 | 0.734 | 0.800 | **0.821** | |
| SEP | 0.705 | 0.753 | 0.863 | 0.757 | 0.769 | 0.788 | 0.812 | **0.842** |

Values on the diagonal (bold) are square of the AVE while the off-diagonals are correlations.

**Table 4. HTMT (Heterotrait–Monotrait ratio).**

|     | ACC | AVO | COL | COM | CPG | SB |
|-----|-----|-----|-----|-----|-----|-----|
| AVO | 0.698 | | | | | |
| COL | 0.748 | 0.765 | | | | |
| COM | 0.803 | 0.768 | 0.851 | | | |
| CPG | 0.780 | 0.776 | 0.848 | 0.890 | | |
| SB  | 0.801 | 0.792 | 0.791 | 0.824 | 0.898 | |
| SEP | 0.785 | 0.827 | 0.811 | 0.852 | 0.873 | 0.883 |

SB (Supervisory behavior), SEP (Sustainable employee performance), ACC (Accommodating), AVO (Avoiding), COL (Collaborating), COM (Compromising), CPG (Competing).

study. Table 3 shows the correlation between latent variables such as the positive correlation between SB and SEP (0.812), SB and CMS (0.843), and CMS and SEP (0.863). Also, the dimensions of CMS individually have positive correlations with SB and SEP, as shown in Table 3. All the correlations are significant at the 0.01 level (Henseler, Ringle, Sarstedt, 2015). An adequate level of discriminant validity was found in the measures used in this study.

## 4.2 Assessment of structural model

*Collinearity issue of structural model*: *VIF* values of all variables were used to determine the collinearity issue of the structural model. *VIF* values are also considered as a reciprocal of tolerance. A standard method bias test based on the *VIF* values was assessed. As scholars [15, 99, 63] suggested that the value of *VIF* equal to or lower than 3.30 is considered biased free. This study shows that all *VIF* values are less than 3.30 (ref Table 5). Therefore, we concluded that the data set was not suffered from a common bias issue.

Moreover, another approach recommended to ensure multicollinearity issues, editors, and scholars require HTMT (Heterotrait-Monotrait) ratio; [100] proposed that the value of constructs should not exceed 0.9. (ref Table 4) which illustrate that the maximum value of a construct found 0.890, henceforth this study is free from multicollinearity issue.

*Coefficient of determination ($R^2$)*: is also called $R^2$ value assessment through PLS-SEM structural model. Researchers argue that $R^2$ represents the independent variance variable by its predictors [101, 102]. Generally, $R^2$ of 0.10 is considerable [103]. However, in PLS-SEM, the $R^2$ value of 0.60 is considered as substantial, 0.33 as moderate, and 0.19 as weak [97]. In this study, the value of $R^2$ 0.769 (ref Table 5) depicts that SB and CMS together cause 76.9% variance in SEP. as per [97], the value falls in substantial influence category.

*The Predictive Relevance ($Q^2$) Effect Sizes*: cross-validated redundancy ($Q^2$) was used in this study to measure the effects of latent variables [104, 105]. The value of $Q^2$ greater than zero is considered as the existence of predictive relevance in the model [106]. The values of $Q^2$ for the present study are presented in Table 5, which is higher than zero. Thus this model has predictive relevance [97].

**Table 5. Saturated model results.**

| Construct | R2 | Adj. R2 | F2 | Q2 | VIF | SRMR |
|-----------|-----|---------|-----|-----|-----|------|
| SEP | 0.769 | 0.765 | 0.476 | 0.498 | 1.000 | 0.073 |

$R^2$ (R-Squared/Coefficient of determination), $F^2$ (The effect size), $Q^2$ (The predictive relevance), VIF (Variance inflation factor), SRMR (Standardized Mean Root Square Residual).

*The effect Sizes $F^2$*: the values of $F^2$ should be higher than 0.02. The present study shows that all the values of $F^2$ are higher than 0.02, which shows there is an effect [63] (ref Table 5).

*Mediation analysis*: to test the mediating role, we used the approach suggested by [92]. According to [107], the main characteristic of indirect effect is that it involves a third variable that plays an intermediate role in the relationship between dependent and independent variables. Technically speaking, the effect of independent variable X on the dependent variable Y is mediated by a third variable M called the mediator. [108], summarized this approach as follows: variable M is a mediator if X significantly account for variability in M, X significantly accounts for variability in Y, M significantly accounts for variability in Y when controlling for X, the effect of X on Y decrease substantially when M is entered simultaneously with X as a predictor of Y. The results illustrate that direct effect from SB to SEP (β = 0.812, *p* = 0.001), and CMS to SEP (β = 0.616, *p* = 0.001) were positive and statistically significant. According to [109] if the direct effect is not significant and indirect effect is significant, full mediation has occurred; if both direct and indirect effects are significant, partial mediation has occurred (refer to Table 8)

**4.2.1 Descriptive statistics.** The descriptive statistics for items and variables are presented in Tables 6 and 7, respectively. 122 observations represent the total number of non-missing values. Range or interval that contains all values in the data, for instance, SB-POS2's range is 3. The minimum and maximum values in a particular item are presented in the table, such as for SB-POS2, 2, and 5, respectively. The sum of the said item is 462. Mean represents the center of the sample observations, such as for SB-POS2, it is 462/122 = 3.786. The higher standard deviation (Std.) values in data show a greater spread in data. Skewness shows the non-symmetrical pattern of the data. When items reach zero, the data becomes more symmetrical. However, negative values show left, and positive values show right skewness. The sample of this study shows left-skewed data. Moreover, kurtosis shows the difference between tails and peaks of a distribution from a normal distribution. A value of zero for kurtosis indicates perfect normal distribution in data such as for SB-TOS1 = 0.044. In this study, the positive kurtosis values exhibit that the distribution has a sharper peak and heavier tails as compared to normal distribution.

## 4.3 Results

**4.3.1 Structural equation modeling.** The standard bootstrapping (500 bootstrap samples) was used with 122 sample observations for the present study to examine the significance of path coefficients [15]. Fig 2 shows the full estimates of the structural equation model, along with the mediating variable of CMS (ref Table 8, Fig 2). As per Table 8, *H1* shows that SB has positive and significant effects on SEP

(β = 0.811, *t* = 22.577, *p* <0.05). *H2* shows significantly positive effects of SB on CMS ((β = 0.841, *t* = 25.193, *p* <0.2). similarly, *H3* shows positive and significant effects of CMS on SEP (β = 0.619, *t* = 5.274, *p* <0.05).

This study also examined the mediating effects of CMS between the relationship of SB, and SEP. the findings show that *H4*: CMS positively and significantly mediates between the relationship of SB and SEP (β = 0.519, *t* = 4.94, *p* <0.05, ref Table 4). Furthermore: this study examined the five dimensions of CMS individually as mediators between the relationship of SB and SEP. the findings (ref Table 8, Fig 2) reveal that *H5a*: *competing* for CMS strategy significantly and positively mediated between the relationship of SB and SEP (β = 0.760, *t* = 18.859, *p* <0.05). *H5b*: *collaborating* CMS strategy positively and significantly mediates between the relationships of SB and SEP (β = 0.758, *t* = 18.368, *p* <0.05). *H5c*: *compromising* CMS strategy significantly and positively mediates between the relationship of SB and SEP

**Table 6. Descriptive statistics for the items.**

| Items | N | Range | Minimum | Maximum | Sum | Mean | | Std. Deviation | Variance | Skewness | | Kurtosis | |
|---|---|---|---|---|---|---|---|---|---|---|---|---|---|
| | Statistic | Statistic | Statistic | Statistic | Statistic | Statistic | Std. Error | Statistic | Statistic | Statistic | Std. Error | Statistic | Std. Error |
| SB-POS1 | 122 | 3 | 2 | 5 | 462 | 3.787 | 0.072 | 0.795 | 0.632 | -0.199 | 0.219 | -0.556 | 0.435 |
| SB-POS2 | 122 | 3 | 2 | 5 | 483 | 3.959 | 0.0835 | 0.922 | 0.85 | -0.191 | 0.219 | -0.243 | 0.435 |
| SB-POS3 | 122 | 4 | 1 | 5 | 468 | 3.836 | 0.0941 | 1.039 | 1.080 | -0.277 | 0.219 | 0.317 | 0.435 |
| SB-TOS1 | 122 | 4 | 1 | 5 | 466 | 3.820 | 0.0798 | 0.882 | 0.777 | -0.12 | 0.219 | 0.244 | 0.435 |
| SB-TOS2 | 122 | 4 | 1 | 5 | 456 | 3.738 | 0.081 | 0.898 | 0.807 | -0.338 | 0.219 | 0.387 | 0.435 |
| SB-TOS3 | 122 | 4 | 1 | 5 | 476 | 3.902 | 0.090 | 0.991 | 0.982 | -0.24 | 0.219 | 0.699 | 0.435 |
| CPG1 | 122 | 4 | 1 | 5 | 489 | 4.008 | 0.085 | 0.940 | 0.884 | -0.387 | 0.219 | 0.714 | 0.435 |
| CPG2 | 122 | 4 | 1 | 5 | 472 | 3.869 | 0.089 | 0.987 | 0.974 | -0.338 | 0.219 | 0.578 | 0.435 |
| CPG3 | 122 | 4 | 1 | 5 | 486 | 3.984 | 0.090 | 0.996 | 0.991 | -0.192 | 0.219 | 0.294 | 0.435 |
| COM1 | 122 | 4 | 1 | 5 | 477 | 3.910 | 0.090 | 0.996 | 0.992 | -0.391 | 0.219 | 0.642 | 0.435 |
| COM2 | 122 | 4 | 1 | 5 | 487 | 3.992 | 0.086 | 0.949 | 0.901 | -0.286 | 0.219 | 0.753 | 0.435 |
| COM3 | 122 | 4 | 1 | 5 | 474 | 3.885 | 0.087 | 0.964 | 0.929 | -0.235 | 0.219 | 0.769 | 0.435 |
| COL1 | 122 | 4 | 1 | 5 | 489 | 4.008 | 0.092 | 1.016 | 1.033 | -0.121 | 0.219 | 0.106 | 0.435 |
| COL2 | 122 | 3 | 2 | 5 | 499 | 4.090 | 0.090 | 0.996 | 0.992 | -0.249 | 0.219 | -0.115 | 0.435 |
| COL3 | 122 | 4 | 1 | 5 | 494 | 4.049 | 0.084 | 0.926 | 0.857 | -0.188 | 0.219 | 0.707 | 0.435 |
| AVO1 | 122 | 4 | 1 | 5 | 465 | 3.812 | 0.086 | 0.948 | 0.898 | -0.208 | 0.219 | 0.094 | 0.435 |
| AVO2 | 122 | 4 | 1 | 5 | 463 | 3.795 | 0.089 | 0.987 | 0.974 | -0.234 | 0.219 | 0.354 | 0.435 |
| AVO3 | 122 | 4 | 1 | 5 | 473 | 3.877 | 0.087 | 0.958 | 0.919 | -0.238 | 0.219 | 0.626 | 0.435 |
| ACC1 | 122 | 4 | 1 | 5 | 477 | 3.910 | 0.092 | 1.012 | 1.025 | -0.134 | 0.219 | 0.639 | 0.435 |
| ACC2 | 122 | 4 | 1 | 5 | 472 | 3.869 | 0.093 | 1.028 | 1.057 | -0.260 | 0.219 | -0.278 | 0.435 |
| ACC3 | 122 | 4 | 1 | 5 | 488 | 4.000 | 0.0839 | 0.927 | 0.86 | -0.186 | 0.219 | 0.686 | 0.435 |
| SEP-C1 | 122 | 3 | 2 | 5 | 483 | 3.959 | 0.0693 | 0.765 | 0.585 | -0.331 | 0.219 | 0.559 | 0.435 |
| SEP-C2 | 122 | 4 | 1 | 5 | 488 | 4.000 | 0.0981 | 1.083 | 1.174 | -0.171 | 0.219 | 0.317 | 0.435 |
| SEP-C3 | 122 | 4 | 1 | 5 | 476 | 3.902 | 0.0843 | 0.931 | 0.866 | -0.290 | 0.219 | 0.742 | 0.435 |
| SEP-C4 | 122 | 3 | 2 | 5 | 480 | 3.934 | 0.0787 | 0.869 | 0.756 | -0.239 | 0.219 | -0.189 | 0.435 |
| SEP-A1 | 122 | 4 | 1 | 5 | 493 | 4.041 | 0.0874 | 0.966 | 0.932 | -0.235 | 0.219 | 0.502 | 0.435 |
| SEP-A2 | 122 | 4 | 1 | 5 | 486 | 3.984 | 0.0815 | 0.900 | 0.81 | -0.236 | 0.219 | 0.694 | 0.435 |
| SEP-A3 | 122 | 3 | 2 | 5 | 493 | 4.041 | 0.081 | 0.894 | 0.800 | -0.127 | 0.219 | 0.357 | 0.435 |
| Valid N (listwise) | 122 | | | | | | | | | | | | |

SB (Supervisory behavior), SEP (Sustainable employee performance), ACC (Accommodating), AVO (Avoiding), COL (Collaborating), COM (Compromising), CPG (Competing), POS (Person-Oriented Skills), TOS (Task-Oriented Skills)

**Table 7. Descriptive statistics for the variables.**

| Items | N | Range | Minimum | Maximum | Sum | Mean | | Std. Deviation | Variance | Skewness | | Kurtosis | |
|---|---|---|---|---|---|---|---|---|---|---|---|---|---|
| | Statistic | Statistic | Statistic | Statistic | Statistic | Statistic | Std. Error | Statistic | Statistic | Statistic | Std. Error | Statistic | Std. Error |
| SB | 122 | 20 | 10 | 30 | 2811 | 23.041 | 0.412 | 4.548 | 20.684 | -0.369 | 0.219 | 0.360 | 0.435 |
| CMS | 122 | 44 | 26 | 70 | 7205 | 59.057 | 1.046 | 11.557 | 133.559 | -0.281 | 0.219 | 0.585 | 0.435 |
| SEP | 122 | 25 | 10 | 35 | 3399 | 27.861 | 0.490 | 5.408 | 29.245 | -0.340 | 0.219 | 0.268 | 0.435 |
| Valid N (listwise) | 122 | | | | | | | | | | | | |

SB (Supervisory behavior), SEP (Sustainable employee performance), CMS (Conflict Management Strategies)

**Table 8. Path coefficients and hypothesis testing.**

| Effect | Relationships | Beta | Mean | (STDEV) | t-value | Decision |
|---|---|---|---|---|---|---|
| *Direct effects* | | | | | | |
| H1 | SB → SEP | 0.812 | 0.811 | 0.036 | 22.577* | Supported |
| H2 | SB → CMS | 0.843 | 0.841 | 0.033 | 25.193* | Supported |
| H3 | CMS → SEP | 0.616 | 0.619 | 0.117 | 5.274* | Supported |
| *Indirect/Mediating effects* | | | | | | |
| H4 | SB → CMS → SEP | 0.519 | 0.522 | 0.105 | 4.94* | Supported |
| H5a | SB → CPG → SEP | 0.760 | 0.758 | 0.040 | 18.859* | Supported |
| H5b | SB → COL → SEP | 0.758 | 0.755 | 0.041 | 18.368* | Supported |
| H5c | SB → COM → SEP | 0.760 | 0.757 | 0.042 | 18.073* | Supported |
| H5d | SB → AVO → SEP | 0.707 | 0.704 | 0.045 | 15.67* | Supported |
| H5e | SB → ACC → SEP | 0.700 | 0.698 | 0.047 | 15.054* | Supported |

Critical value *p<0.05; SB (Supervisory behavior), SEP (Sustainable employee performance), ACC (Accommodating), AVO (Avoiding), COL (Collaborating), COM (Compromising), CPG (Competing).

($\beta$ = 0.760, $t$ = 18.073, $p$ <0.05). *H5d*: *avoiding* CMS strategy also positively and significantly mediates between the relationship of SB and SEP ($\beta$ = 0.707, $t$ = 15.67, $p$ <0.05). Lastly, *H5e*: accommodating CMS strategy significantly and positively mediates between the relationship of SB and SEP ($\beta$ = 0.700, $t$ = 15.054, $p$ <0.05). Hence, supporting all the mediating hypothesis (*H4, H5a, H5b, H5c, H5d, and H5e*).

## 5. Conclusion and discussion

The present study has achieved its overall goals and validates all the hypotheses. Hypothesis 1 and 2 examine that SB positively and significantly affects SEP and CMS. The study revealed

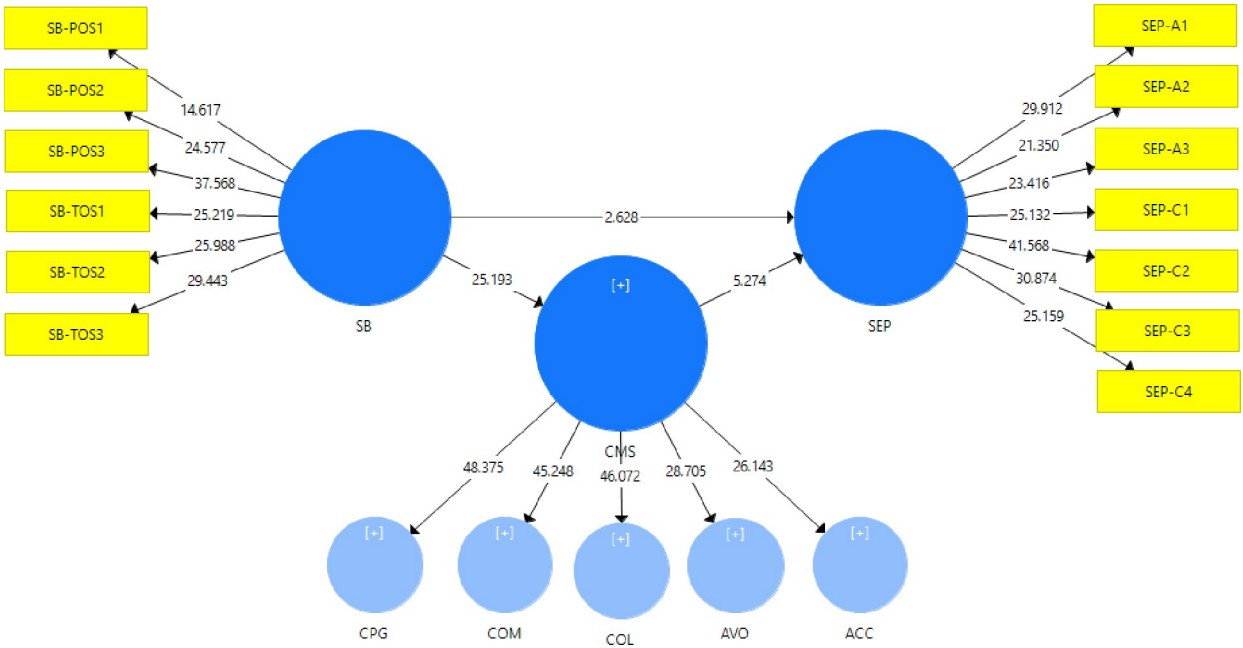

**Fig 2. Partial least square SEM model (shows positive relation between the proposed model).**

that both person-oriented skills and task-oriented skills of the supervisors have significant impacts concerning SEP and CMS. Person oriented skills of SB such as recognizing and rewarding good performance, willingness to listen to employees' problems, and treating employees with respect affect SEP positively. Also, the task-oriented skills of a supervisor, such as setting specific goals for employees, emphasizing high standards of performance for individual employees and team, significantly affect the SEP and CMS [90, 89]. The findings of the study revealed that SB and CMS positively predict employees' performance under SEP. The performance indicators such as maintenance of performance at work, acceptance, learning through feedback, cooperation, effective communication, showing resiliency, creative thinking, and keeping job knowledge up to date, were positively affected by SB and CMS [91]. The results of hypothesis 1 (SP positively affects SEP) were supported by past research [24, 7, 3, 6]. Also, the results of hypothesis 2 (SB positively affects CMS) were supported by past research [19, 22, 21].

CMS plays a significant role in solving or lowering down the negative impacts of conflicts at the workplace. This study examined that competing, collaborating, compromising, avoiding, and accommodating strategies play a vital role in conflict resolution [19]. This study found that CMS positively and significantly affects the SEP (H4), and the results were supported by many studies such as A. [18, 11, 19, 72, 16, 17]. Furthermore, the present study shows the complexity of CMS as a mediator between the relationships of SB and SEP. The dimensions of CMS positively and significantly mediated the relationship between SB and SEP. Past research has supported the results of the present study that all the dimensions of the CMS play a significant role under particular situations. All the mediation hypotheses were supported by past studies such as [110, 73, 19, 75, 76, 74]. The use of CMS strategies profoundly depends on the country, context, and magnitude of the conflicts. For instance, [110, 111] found integrating and avoiding styles of CMS negatively and compromising, dominating, and obliging conflict management styles correlated with counterproductive work behaviors within Ghanaian financial, telecommunication, manufacturing, and computer software SMEs. Similarly, [18] found that integrating strategy positively affects employees' job satisfaction and turnover intention on Chinese SMS. Finally, this study focuses on CMS strategies, as suggested by Kodikal et al.

## 5.1 Theoretical implications

This study validates the definite link between SB, CMS, and SEP. Serval theoretical perspectives, including mediation of the chosen variables, were proven in this study. Results show that CMS is a positive mediator. First, SB needs to be altered according to using CMS to enhance SEP; this proves CMS as a positive mediator and SB and CMS together as a significant predictor of SEP. Secondly, this study found that competing, collaborating, compromising, accommodating and avoiding strategies of conflict management mediate between the relationship of SB and SEP. This empirical study, unlike other studies, found the critical mediating relationship of five CMS between SB and SEP.

## 5.2 Practical and managerial implications

The present study is of enormous importance for SMEs for dealing with conflicts at the workplace and choosing the right CMS for particular conflicts. The CMS strategies will help organizations to enhance SEP. Supervisors and managers at SMEs can make strategies and choose the best suitable CMS for each situation to optimize the employees' performance ultimately. This study highlights the possible CMS such as competing, collaborating, compromising, accommodating, and avoiding for supervisors to understand and apply them accordingly to predict SEP. Similarly, higher management of SMEs can consider the outcomes of this study

while developing strategic CMS and policies. Furthermore, supervisors can practice the person-oriented and task-oriented skills presented in the "conclusion and discussion" section of this study to successfully adopt and apply suitable CMS in their organizations.

### 5.3 Limitations and future research directions

There were few limitations associated with the present study. The time limit bounded to obtain the maximum number of responses. Also, lack of financial resources bounded to collect responses only from the SMEs situated in twin cities Rawalpindi and Islamabad; hence, it is difficult to offer generalizability in this study. Consequently, future research may focus on collecting data from SMEs in other cities of Pakistan to offer a better generalization of findings. The study mainly collected data from manufacturing SMEs randomly; the future studies can target other industries as well. Moreover, this study focused on cross-sectional data; we suggest to develop more in-depth knowledge through longitudinal studies and to investigate relationships between the constructs. Also, to find the differences in the results of cross-sectional and longitudinal studies. This study collected data from 122 employees of SMEs; future studies can increase the sample size to get more diverse responses. Similarly, this study collected data from employees of SMEs; further studies can explore multiple relationships of the model between different hierarchal levels such as executives, middle-level managers, and workers. Different cultural contexts and different approaches to treating the data could diversify future studies.

## Supporting information

**S1 Data.**
(DOC)

**S2 Data.**
(CSV)

**S1 Table.**
(DOCX)

## Author Contributions

**Conceptualization:** Jiang Min, Shuja Iqbal, Muhammad Aamir Shafique Khan, Shamim Akhtar, Farooq Anwar.

**Data curation:** Jiang Min, Shuja Iqbal, Muhammad Aamir Shafique Khan, Shamim Akhtar, Farooq Anwar.

**Formal analysis:** Jiang Min, Shuja Iqbal, Muhammad Aamir Shafique Khan, Shamim Akhtar, Farooq Anwar, Sikandar Ali Qalati.

**Investigation:** Jiang Min, Shuja Iqbal, Muhammad Aamir Shafique Khan, Shamim Akhtar, Farooq Anwar.

**Methodology:** Jiang Min, Shuja Iqbal, Muhammad Aamir Shafique Khan, Shamim Akhtar, Farooq Anwar.

**Project administration:** Jiang Min, Shuja Iqbal, Muhammad Aamir Shafique Khan, Shamim Akhtar, Farooq Anwar.

**Resources:** Jiang Min, Shuja Iqbal, Muhammad Aamir Shafique Khan, Shamim Akhtar, Farooq Anwar.

**Software:** Jiang Min, Shuja Iqbal, Muhammad Aamir Shafique Khan, Shamim Akhtar, Farooq Anwar, Sikandar Ali Qalati.

**Supervision:** Jiang Min, Shuja Iqbal, Muhammad Aamir Shafique Khan, Shamim Akhtar, Farooq Anwar.

**Validation:** Jiang Min, Shuja Iqbal, Muhammad Aamir Shafique Khan, Shamim Akhtar, Farooq Anwar.

**Visualization:** Jiang Min, Shuja Iqbal, Muhammad Aamir Shafique Khan, Shamim Akhtar, Farooq Anwar.

**Writing – original draft:** Jiang Min, Shuja Iqbal, Muhammad Aamir Shafique Khan, Shamim Akhtar, Farooq Anwar.

**Writing – review & editing:** Jiang Min, Shuja Iqbal, Muhammad Aamir Shafique Khan, Shamim Akhtar, Farooq Anwar, Sikandar Ali Qalati.

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
