## [Editor Report · Decision Letter 0]

17 Feb 2020

PONE-D-20-01676

Impact of Supervisory Behavior on Sustainable Employee Performance: Mediation of Conflict Management Strategies

PLOS ONE

Dear Dr. KHAN,

Thank you for submitting your manuscript to PLOS ONE. After careful consideration, we feel that it has merit but does not fully meet PLOS ONE’s publication criteria as it currently stands. Therefore, we invite you to submit a revised version of the manuscript that addresses the points raised during the review process.

We would appreciate receiving your revised manuscript by Apr 02 2020 11:59PM. To enhance the reproducibility of your results, we recommend that if applicable you deposit your laboratory protocols in protocols.io, where a protocol can be assigned its own identifier (DOI) such that it can be cited independently in the future. For instructions see: http://journals.plos.org/plosone/s/submission-guidelines#loc-laboratory-protocols

We look forward to receiving your revised manuscript.

Kind regards,

Dejan Dragan, PhD

Academic Editor

PLOS ONE

1. Please improving statistical reporting and refer to p-values as "p<.001" instead of "p=.000". Our statistical reporting guidelines are available at https://journals.plos.org/plosone/s/submission-guidelines#loc-statistical-reporting

2. If the language of the questionnaire was other than English, please also include a copy of it the original language as Supporting Information.

3. Please ensure that you refer to Figure 1 in your text as, if accepted, production will need this reference to link the reader to the figure.

Additional Editor Comments (if provided):

Editor’s comments to the paper:

Impact of Supervisory Behavior on Sustainable Employee Performance: Mediation of Conflict Management Strategies

The paper investigates the relationship between supervisory behavior, conflict management strategies, and sustainable employee performance. Additionally, the study has examined the mediating effects of conflict management strategies between supervisory behavior and sustainable employee performance relationships. The significance of the model was assessed using the PLS-SEM. The paper adds in the current literature of being supervisory behavior as a critical predictor of sustainable employee performance. Additionally, the study adds in the current literature of PLS-SEM an assessment model for direct and mediation relationships. The study seems interesting and useful in the corresponding research field.

The paper, in general, in rough terms satisfies major rigor requirements that are demanded from Plos One. The red path remains relatively consistent throughout the paper; the main contributions and findings are clearly enough emphasized, analyzed, and justified. Moreover, in general, the paper is in most places, more or less sufficiently organized and written. So, to summarize, the contribution of the submitted paper seems relevant and useful for the researchers from the field.

However, the editor has detected some major issues that are recommended to be corrected prior to giving the paper to the reviewers and carrying out a further publishing process. Some major comments are:

1. The editorial board has expressed a certain doubt whether the paper meets PLOS ONE criteria for papers considering whether the limitations of observational studies have been acknowledged, including in the abstract; whether there are unsupported statements of causation; and whether the analysis is affected by confounding variables, a lack of generalizability, selective reporting, post hoc analyses, or data dredging. The editor thus appeals to the authors to check and reconsider this issue.

2. In the title, the term PLS-SEM is recommended to be included.

3. Please use the “structural equation” instead of “structured equation”.

4. The aesthetic look and resolution of figure 2 should be improved. The title of this figure is inadequate. Please use “The derived PLS-SEM model” or something similar. Denote the statistical significance of the parameters in Figure 2.

5. Also, it should be explained more clearly why the PLS-SEM modeling was used instead of the Covariance-based SEM (in most cases, the main reason should be a prediction – please look at Hair, Chong (2016): An updated and expanded assessment of PLS-SEM in information systems research).

6. In the literature review, the brief explanation of structural equation modeling and PLS-SEM methodology is missing, i.e., several sentences should be devoted to the methodologies used in the field.

7. Although it is relatively clear enough emphasized what the main contribution of the paper is, i.e., what has been done new, it is maybe not clear enough highlighted what are the main differences, if the novelties in this paper are compared with the newest state-of-the-art. Here, perhaps more precise borderline should be more clearly highlighted, such as for example, that this kind of research is a first attempt of “…”, which has not been designed or detected in the field yet. So, to summarize, please, clearly point to the research gap (in the introduction or literature review), that has been targeted by this paper, and mention where, how, and to what extent similar studies have been conducted, with the precise border between this study, and the other studies.

8. When introducing the hypotheses, it might be perhaps convenient to put them into some transparent, nice-looking table at the end of their explanation.

9. Before the appearance of Figure 1, please immediately and clearly define symbols of all variables (i.e., the original indicator variables and the latent factors) with adequate mathematical rigor. To do this, I recommend the Mathtype (please use it for all cases when presenting or mentioning any of the variables throughout the paper). In figure 1, please include all clusters of indicators in the sense that the nature of variables and causal paths among constructs and items are immediately evident (formative, reflective?).

10. In the methodological section, I miss a short description of structural equation modeling in general sense, as well as the PLS-SEM methodology in a specific sense.

11. When explaining the details of the survey, please include a brief, compact table of its properties and characteristics.

12. I am missing some of the basic descriptive statistics for the items, i.e., the mean, std, skewness, kurtosis,.., which would more clearly show the character and the level of non-normality of the data.

13. Please insert line numbers throughout all the paper.

14. Regarding all issues about statistical modeling and reporting of statistical results, the authors are invited to synchronize their work with the guidelines and rules of thumb of the afore-mentioned paper:

1. Hair, Chong (2016): An updated and expanded assessment of PLS-SEM in information systems research.

Moreover, it would be appropriate to look at the following papers as well:

2. Jörg Henseler & Christian M. Ringle & Marko Sarstedt (2015): A new criterion for assessing discriminant validity in variance-based structural equation modeling;

3. Dijkstra, Henseler (2015): Consistent and asymptotically normal PLS estimators for

linear structural equations, and

4. Dijkstra, Henseler (2015): Consistent Partial Least Squares Path Modeling.

At this very moment, there are several issues detected in the present paper, that are not synchronized with the newest findings of the just-mentioned papers of leading World’s experts regarding the SEM modeling. For instance, just to mention some issues:

AD1. According to the paper: A new criterion for assessing discriminant validity in variance-based structural equation modeling, authors Jörg Henseler & Christian M. Ringle & Marko Sarstedt (2015), the “…the heterotrait-monotrait ratio (HTMT) of the correlations, which is the average of the heterotrait-heteromethod correlations (i.e., the correlations of indicators across constructs measuring different phenomena), relative to the average of the monotrait-heteromethod correlations (i.e., the correlations of indicators within the same construct)….” should be concerned when a discriminant validity is tried to be accessed. The authors of this paper also emphasize: “…Fornell-Larcker criterion and the assessment of the cross-loadings fail to reliably uncover discriminant validity problems in variance-based SEM…”. Please, take into account this issue, i.e., include the HTMT besides the Fornell-Larcker criterion and the estimated cross-loadings.

AD2. In the Hair, Chong (2016) paper, there is a table I. regarding the differences between PLS-SEM and CB-SEM about rules of thumb for choosing the SEM method:

Thus, the major rules of thumb for choosing the SEM method are:

1. The research objective is exploratory or confirmation of theory based on total variance.

2. The objective of the analysis is a prediction.

3. The measurement philosophy is estimation with the composite factor model using a total variance.

4. The research objective is to explain the relationships between exogenous and endogenous constructs.

5. The structural and/or measurement models are complex (many constructs and many indicators).

6. Formatively measured constructs are specified in the research.

7. The preferred method when the sample size is small (about 100 samples). But PLS is also an excellent method for larger samples.

8. The data are not normally distributed.

9. The scaling of responses is ordinal or nominal.

10. The data is secondary/archival, particularly single-item measures.

11. The research objective is to use latent variable scores in subsequent analyses.

12. The structural model will be estimated with a higher-order construct that has only two first-order constructs.

13. The analysis involves a continuous moderator.

14. The investigation will examine the model for unobserved heterogeneity

Be cautious that all points emphasized above are consistent with the research and reporting in this paper.

AD3. In the Hair, Chong (2016) paper, there is a sub-section Critical issues in PLS-SEM IS applications regarding the eight critical issues that must be conducted in the research and appropriately reported:

• reasons are given for using PLS-SEM,

• model descriptive statistics,

• sampling characteristics,

• technical reporting,

• formative measurement metrics,

• reflective measurement metrics,

• structural model metrics, and

• additional analyses such as mediation, moderation, multi-group analyses, and common methods variance.

AD4. In the Hair, Chong (2016) paper, the following paragraph is particularly important and should be perhaps briefly mentioned in the paper:

The latent variable scores for CB-SEM are indeterminant – i.e., an infinite number of different sets of latent variable scores that will fit the model equally well are possible for a CB-SEM solution, which makes CB-SEM unsuitable for prediction (Hair et al., 2016, 2018). In contrast, the PLS-SEM method always produces a single determinant score for each SEM composite for each observation. Moreover, in CB-SEM prediction, the R2 is related to the proportion of common variance explained, whereas in PLS-SEM, the R2 is related to the proportion of total variance explained (Hair et al., 2018). Thus, PLS-SEM is always the preferred SEM method when the research objective is prediction and it is believed that this reason for selecting PLS-SEM rather than CB-SEM will increase considerably in the future.

AD5: In the Hair, Chong (2016) paper, there is also Table X explaining the best practices about reporting the PLS-SEM results. This table should be at the top of the list of concerns that the authors should be cautious with the most significant possible attention. The snapshot of this table is as following:

The authors are invited that (at least) in rough terms take into consideration these issues from Table X, particularly regarding reporting the results of the final model (Besides R2 and t values of the estimated coefficients, Predictive relevance (Q2), and p values of the estimated coefficients’ significance), maybe also some other measures from Table X and/or those measures that are offered from the Smart-PLS documentation (https://www.smartpls.com/documentation/algorithms-and-techniques/model-fit) might have been perhaps included in the paper. Accordingly, the authors are invited to reconsider this issue, whether it is also appropriate to include some other, additional measures.

15. The used software must always be reported (Smart-PLS?).
---

## [Author Response · Author response to Decision Letter 0]

9 Apr 2020

Editor’s comments to the paper:

Impact of Supervisory Behavior on Sustainable Employee Performance: Mediation of Conflict Management Strategies

The paper investigates the relationship between supervisory behavior, conflict management strategies, and sustainable employee performance. Additionally, the study has examined the mediating effects of conflict management strategies between supervisory behavior and sustainable employee performance relationships. The significance of the model was assessed using the PLS-SEM. The paper adds in the current literature of being supervisory behavior as a critical predictor of sustainable employee performance. Additionally, the study adds in the current literature of PLS-SEM an assessment model for direct and mediation relationships. The study seems interesting and useful in the corresponding research field.

The paper, in general, in rough terms satisfies major rigor requirements that are demanded from Plos One. The red path remains relatively consistent throughout the paper; the main contributions and findings are clearly enough emphasized, analyzed, and justified. Moreover, in general, the paper is in most places, more or less sufficiently organized and written. So, to summarize, the contribution of the submitted paper seems relevant and useful for the researchers from the field.

However, the editor has detected some major issues that are recommended to be corrected prior to giving the paper to the reviewers and carrying out a further publishing process. Some major comments are:

1. The editorial board has expressed a certain doubt whether the paper meets PLOS ONE criteria for papers considering whether the limitations of observational studies have been acknowledged, including in the abstract; whether there are unsupported statements of causation; and whether the analysis is affected by confounding variables, a lack of generalizability, selective reporting, post hoc analyses, or data dredging. The editor thus appeals to the authors to check and reconsider this issue. 

We have included limitations of observational studies in limitations section and in the abstract as well. 

Limitations:

There were a few limitations associated with the present study. The time constraint was there to collect more responses. Also, the financial aspects forced to collect responses only from the SMEs situated in twin cities Rawalpindi and Islamabad; hence, it is difficult to offer generalizability in this study. Consequently, future research may focus on collecting data from SMEs in other cities of Pakistan to offer better generalization of findings. The study mainly collected data from manufacturing SMEs randomly; the future studies can target other industries as well. Moreover, this study focused on cross-sectional data, we suggest to develop more in-depth knowledge through longitudinal studies, and to investigate relationships between the constructs. Also, to find the differences in the results of cross-sectional and longitudinal studies. This study collected data from 122 employees of SMEs; future studies can increase the sample size to get more diverse responses. Similarly, this study collected data from employees of SMEs; further studies can explore multiple relationships of the model between different hierarchal levels such as executives, middle-level managers, and workers. Different cultural contexts and different approaches to treating the data could diversify future studies. (Lines 526-39)

Abstract: 

This study is based on cross-sectional data, more longitudinal studies can explore relationships between the constructs. (Line 43-45)

In this study we have tried to discuss any relationships, specifically to the constructs supported by the past studies in hypothesis development section. 

We have stated the generalizability issue in the limitations section of the paper as follows:

Also, the financial aspects forced to collect responses only from the SMEs situated in twin cities Rawalpindi and Islamabad; hence, it is difficult to offer generalizability in this study. Consequently, future research may focus on collecting data from SMEs in other cities of Pakistan to offer better generalization of findings. (Lines 527-30)

We have tried in this study to avoid selective reporting by searching multiple databases to find relevant studies and analyze them in relations to our study. We have tried to discuss the proposed hypothesis in relation to past studies to provide an initial bases of prediction, followed by the data analysis to generate results. Multiple tests in SmartsPLS were conducted to evaluate the reliability and validity of data, and model fit were explained. 

2. In the title, the term PLS-SEM is recommended to be included.

The revised title as follows:

Impact of Supervisory Behavior on Sustainable Employee Performance: Mediation of Conflict Management Strategies using PLS-SEM (Lines 1-3). 

3. Please use the “structural equation” instead of “structured equation.”

We have revised “structured equation” to “structural equation” throughout the paper. 

4. The aesthetic look and resolution of figure 2 should be improved. The title of this figure is inadequate. Please use “The derived PLS-SEM model” or something similar. Denote the statistical significance of the parameters in Figure 2.

We have revised the Figure 2’s quality and title as follows:

Figure 2: The derived PLS-SEM model

5. Also, it should be explained more clearly why the PLS-SEM modeling was used instead of the Covariance-based SEM (in most cases, the main reason should be a prediction – please look at Hair, Chong (2016): An updated and expanded assessment of PLS-SEM in information systems research).

We have discussed it in the literature review section of the paper as follows: 

This study uses SEM as a standard reporting method to allow replicability and establish rigor. Partial least square structural equation model (PLS-SEM) has been used in different disciplines such as strategic management and marketing (Hair, Sarstedt, Ringle and Mena, 2012), operation and international management (Peng and Lai, 2012; Richter et al, 2016), accounting (Lee et al., 2011), tourism (do-Valle and Assaker, 2015), family business (Sarstedt et al., 2014), organization and group research (Sosik et al., 2009). Moreover, this study used PLS-SEM instead of covariance-based SEM to predict the dependent affected by latent variables. Furthermore, an increasing trend was seen in published articles using PLS-SEM (Hair and Chong, 2017). The rationale for the choice of PLS-SEM was as follows. First, in CB-SEM the scores of latent variables are indeterminant, which makes it unsuitable to use in predictive studies. In comparison, PLS-SEM produces “a single determinant score for each SEM composite for each observation” (Hair et al., 2018). Secondly, in CB-SEM R2 relates to the proportion of common variance explained. In contrast in PLS-SEM R2 relates to the total variance explained (Hair et al., 2018). (Lines 175-83). 

6. In the literature review, the brief explanation of structural equation modeling and PLS-SEM methodology is missing, i.e., several sentences should be devoted to the methodologies used in the field. 

We have discussed the PLS-SEM methodology in literature review and methodology section. (Lines 175-88 and 329-35). 

7. Although it is relatively clear enough emphasized what the main contribution of the paper is, i.e., what has been done new, it is maybe not clear enough highlighted what are the main differences, if the novelties in this paper are compared with the newest state-of-the-art. Here, perhaps more precise borderline should be more clearly highlighted, such as for example, that this kind of research is a first attempt of “…”, which has not been designed or detected in the field yet. So, to summarize, please, clearly point to the research gap (in the introduction or literature review), that has been targeted by this paper, and mention where, how, and to what extent similar studies have been conducted, with the precise border between this study, and the other studies.

We have revised the contribution of the study as follows: 

This study contributes as follows. We used a more inclusive discernment to explore the multifaceted mediation role of CMS on SEP. Past studies on CMS mostly examined its direct effects on constructs such as efficacy and performance (Alper et al., 2000), innovation performance (Song and Lim, 2006), interpersonal rewards and performance (Weider-Hatfield and Hatfield, 1996), job satisfaction, cognition of action taking and sleep disorder (Way, Jimmieson, and Bordia, 2014). Moreover, previous studies have found positive relationship of SB and CMS ((Kodikal et al., 2014; Thomas, 1992; Zhao, Thatcher, and Jehn, 2019; Sabanci, Sahin, and Özdemir, 2016), employee performance and SEP (Kohli, 1989; Gilbreath and Benson, 2004; Fatima and Azam, 2016; Jiang et al., 2017; Ellinger, Ellinger, and Keller, 2003). However, this study uniquely examines the mediation role of CMS between the relationship of SB and SEP. Finally, this study enriches the literature on SB, CMS and SEP and provides substantial practical implications for managers at SMEs to enhance SEP. (Lines 86-97)

8. When introducing the hypotheses, it might be perhaps convenient to put them into some transparent, nice-looking table at the end of their explanation.

we have added a table exhibiting all hypothesis. (Annexure 1, lines 763-64)

9. Before the appearance of Figure 1, please immediately and clearly define symbols of all variables (i.e., the original indicator variables and the latent factors) with adequate mathematical rigor. To do this, I recommend the Mathtype (please use it for all cases when presenting or mentioning any of the variables throughout the paper). In figure 1, please include all clusters of indicators in the sense that the nature of variables and causal paths among constructs and items are immediately evident (formative, reflective?).

Suggested changes were made throughout the manuscript including Figure 1. 

10. In the methodological section, I miss a short description of structural equation modeling in general sense, as well as the PLS-SEM methodology in a specific sense.

This study used partial least square (PLS) modeling to analyze the conceptual model. We used PLS path modeling because it has received waste application in management and related fields (Hairet al., 2012; Kura, 2016; Kura et al., 2015). This study aimed to predict the dependant variable, hence PLS path modeling was considere a suitable investigative method (Hair et al., 2011). Scholars suggests PLS as “most fully developed and general system” (McDonald, 1996) concerning the “variance based structured equation modeling” method (Ringle et al., 2015). Therefore, the data were further analyzed using Smart-PLS 3 to examine the proposed relationships. (Lines 329-35). 

11. When explaining the details of the survey, please include a brief, compact table of its properties and characteristics.

We have added a table explaining details of survey (demographic information of repondents). (Lines 326-28). 

12. I am missing some of the basic descriptive statistics for the items, i.e., the mean, std, skewness, kurtosis,.., which would more clearly show the character and the level of non-normality of the data.

We have added the basic descriptive statistics for the items and variables. (Lines 421-37). 

13. Please insert line numbers throughout all the paper.

We have added line numbers throughout all the paper. 

14. Regarding all issues about statistical modeling and reporting of statistical results, the authors are invited to synchronize their work with the guidelines and rules of thumb of the afore-mentioned paper:

1. Hair, Chong (2016): An updated and expanded assessment of PLS-SEM in information systems research. 

Moreover, it would be appropriate to look at the following papers as well:

2. Jörg Henseler & Christian M. Ringle & Marko Sarstedt (2015): A new criterion for assessing discriminant validity in variance-based structural equation modeling;

3. Dijkstra, Henseler (2015): Consistent and asymptotically normal PLS estimators for

linear structural equations, and

4. Dijkstra, Henseler (2015): Consistent Partial Least Squares Path Modeling.

At this very moment, there are several issues detected in the present paper, that are not synchronized with the newest findings of the just-mentioned papers of leading World’s experts regarding the SEM modeling. For instance, just to mention some issues:

AD1. According to the paper: A new criterion for assessing discriminant validity in variance-based structural equation modeling, authors Jörg Henseler & Christian M. Ringle & Marko Sarstedt (2015), the “…the heterotrait-monotrait ratio (HTMT) of the correlations, which is the average of the heterotrait-heteromethod correlations (i.e., the correlations of indicators across constructs measuring different phenomena), relative to the average of the monotrait-heteromethod correlations (i.e., the correlations of indicators within the same construct)….” should be concerned when a discriminant validity is tried to be accessed. The authors of this paper also emphasize: “…Fornell-Larcker criterion and the assessment of the cross-loadings fail to reliably uncover discriminant validity problems in variance-based SEM…”. Please, take into account this issue, i.e., include the HTMT besides the Fornell-Larcker criterion and the estimated cross-loadings.

We have inculded the HTMT ration in the paper and consulted the suggested papers to meet the rule of thumb. 

Scholars Gold et al. (2001), Teo et al. (2008), Hair and Chong (2017) suggests to measure the multicollinearity in data by HTMT ratio, which should not be higher than 0.9. This study met the threshold exhibited in Table 3. (Lines 383-85 and 388-89). 

Furthermore, we have carefully read the suggested paper and synchronized all of the results of the paper with suggested studies. 

AD2. In the Hair, Chong (2016) paper, there is a table I. regarding the differences between PLS-SEM and CB-SEM about rules of thumb for choosing the SEM method:

Thus, the major rules of thumb for choosing the SEM method are:

1. The research objective is exploratory or confirmation of theory based on total variance.

2. The objective of the analysis is a prediction. 

3. The measurement philosophy is estimation with the composite factor model using a total variance.

4. The research objective is to explain the relationships between exogenous and endogenous constructs.

5. The structural and/or measurement models are complex (many constructs and many indicators).

6. Formatively measured constructs are specified in the research.

7. The preferred method when the sample size is small (about 100 samples). But PLS is also an excellent method for larger samples.

8. The data are not normally distributed.

9. The scaling of responses is ordinal or nominal.

10. The data is secondary/archival, particularly single-item measures.

11. The research objective is to use latent variable scores in subsequent analyses.

12. The structural model will be estimated with a higher-order construct that has only two first-order constructs.

13. The analysis involves a continuous moderator.

14. The investigation will examine the model for unobserved heterogeneity

Be cautious that all points emphasized above are consistent with the research and reporting in this paper.

We have craefully revised and made sure that the choice of PLS-SEM meets the rule of thumb in our study. 

AD3. In the Hair, Chong (2016) paper, there is a sub-section Critical issues in PLS-SEM IS applications regarding the eight critical issues that must be conducted in the research and appropriately reported:

• reasons are given for using PLS-SEM, 

• model descriptive statistics, 

• sampling characteristics, 

• technical reporting, 

• formative measurement metrics, 

• reflective measurement metrics, 

• structural model metrics, and 

• additional analyses such as mediation, moderation, multi-group analyses, and common methods variance.

We have carefully mentioned above points in the manuscript. 

AD4. In the Hair, Chong (2016) paper, the following paragraph is particularly important and should be perhaps briefly mentioned in the paper:

The latent variable scores for CB-SEM are indeterminant – i.e., an infinite number of different sets of latent variable scores that will fit the model equally well are possible for a CB-SEM solution, which makes CB-SEM unsuitable for prediction (Hair et al., 2016, 2018). In contrast, the PLS-SEM method always produces a single determinant score for each SEM composite for each observation. Moreover, in CB-SEM prediction, the R2 is related to the proportion of common variance explained, whereas in PLS-SEM, the R2 is related to the proportion of total variance explained (Hair et al., 2018). Thus, PLS-SEM is always the preferred SEM method when the research objective is prediction and it is believed that this reason for selecting PLS-SEM rather than CB-SEM will increase considerably in the future.

We have discussed the suggested paragraph in the manuscript as follows: 

The rationale for the choice of PLS-SEM was as follows. First, in CB-SEM the scores of latent variables are indeterminant, which makes it unsuitable to use in predictive studies. In comparison, PLS-SEM produces “a single determinant score for each SEM composite for each observation” (Hair et al., 2018). Secondly, in CB-SEM R2 relates to the proportion of common variance explained. In contrast in PLS-SEM R2 relates to the total variance explained (Hair et al., 2018). (Lines 183-88). 

AD5: In the Hair, Chong (2016) paper, there is also Table X explaining the best practices about reporting the PLS-SEM results. This table should be at the top of the list of concerns that the authors should be cautious with the most significant possible attention. The snapshot of this table is as following:

The authors are invited that (at least) in rough terms take into consideration these issues from Table X, particularly regarding reporting the results of the final model (Besides R2 and t values of the estimated coefficients, Predictive relevance (Q2), and p values of the estimated coefficients’ significance), maybe also some other measures from Table X and/or those measures that are offered from the Smart-PLS documentation (https://www.smartpls.com/documentation/algorithms-and-techniques/model-fit) might have been perhaps included in the paper. Accordingly, the authors are invited to reconsider this issue, whether it is also appropriate to include some other, additional measures. 

All of the suggested tests in the table were duly reported in the manuscript. 

15. The used software must always be reported (Smart-PLS?).

We have reported the software used as follows: 

The data were further analyzed using Smart-PLS 3 to examine the proposed relationships. (Lines 334-35).

---

## [Decision Letter · Decision Letter 1]

21 May 2020

PONE-D-20-01676R1

Impact of Supervisory Behavior on Sustainable Employee Performance: Mediation of Conflict Management Strategies using PLS-SEM

PLOS ONE

Dear Authors,

Thank you for submitting your manuscript to PLOS ONE. After careful consideration, we feel that it has merit but does not fully meet PLOS ONE’s publication criteria as it currently stands. Therefore, we invite you to submit a revised version of the manuscript that addresses the points raised during the review process.

We look forward to receiving your revised manuscript.

Kind regards,

Dejan Dragan, PhD

Academic Editor

PLOS ONE

Additional Editor Comments (if provided):

The two reviewers have evaluated that the paper is suitable to be accepted, while the third one requires the major revision. I suggest that the authors carefully follow the instructions of the reviewer #1 to increase the likelihood of acceptance of the paper. The AE.

Reviewers' comments:

Reviewer's Responses to Questions

**Comments to the Author**

1. If the authors have adequately addressed your comments raised in a previous round of review and you feel that this manuscript is now acceptable for publication, you may indicate that here to bypass the “Comments to the Author” section, enter your conflict of interest statement in the “Confidential to Editor” section, and submit your "Accept" recommendation.

Reviewer #1: (No Response)

Reviewer #2: All comments have been addressed

Reviewer #3: All comments have been addressed

2. Is the manuscript technically sound, and do the data support the conclusions?

Reviewer #1: Partly

Reviewer #2: Yes

Reviewer #3: Yes

3. Has the statistical analysis been performed appropriately and rigorously? 

Reviewer #1: No

Reviewer #2: Yes

Reviewer #3: Yes

4. Have the authors made all data underlying the findings in their manuscript fully available?

Reviewer #1: No

Reviewer #2: No

Reviewer #3: Yes

5. Is the manuscript presented in an intelligible fashion and written in standard English?

Reviewer #1: Yes

Reviewer #2: Yes

Reviewer #3: Yes

6. Review Comments to the Author

Reviewer #1: REVIEW REPORT

Strengths

The enthusiasm of the authors towards the topic is seen throughout the manuscript. The Manuscript is also detailed and contributes immensely to literature on managing conflicts among employees in SMEs. The authors succeeded in identifying the relevance of their research. However, I found the writing and organization of the manuscript uneven. There are problems in all the sections, including your mediation analysis.

Weaknesses

ABSTRACT

1. The Abstract will benefit from English Language editing. You mixed up the tenses in the Abstract. For instance, in lines 32 to 34, the purpose…is to investigate…Additionally, the study has examined…Please improve the language.

2. In line 38 “relationship”, needs correction. Lines 43 and 44 need correction. The study…more longitudinal…constructs. Line 45 needs correction. It should read as “…the study adds to the current…

INTRODUCTION

1. The Introduction will benefit from English Language editing. For instance, check the preposition after “add”. Check throughout for this problem.

2. In line 71, change the word “primitively” to a more suitable synonym.

3. In line 87, is the mediation role of CMS on SEP, or on the relationship between SB and SEP?

4. Restructure sentence in line 86.

5. Authors should provide brief information on PLS-SEM in the introduction.

LITERATURE REVIEW

1. The study has more than one hypothesis, so correct the spelling in line 98. Check throughout for this problem.

2. In lines 154 and 155 … brake down, and dysfunctional… need correction.

3. Line 159, you CANNOT start a sentence with a citation: (De-Dreu et al., 1999). Write the name of the author(s), then add the citation. Check throughout for this problem.

4. Adding the review on the PLS-SEM to the review on the CMS is not right. PLS-SEM and CMS are different. Authors should have a separate sub-heading for PLS-SEM.

5. This: “In the literature review, the brief explanation of structural equation modeling and PLS-SEM methodology is missing, i.e., several sentences should be devoted to the methodologies used in the field” should be included as indicated earlier. Authors only showed the reason for using PLS-SEM. You should provide a review of structural equation modeling as well as PLS-SEM methodology.

HYPOTHESES DEVELOPMENT

1. Line 194…concerning to… need correction. The sentence in line 196 needs correction. The sentence in lines 203 to 205 needs correction. Line 209, change “ample” to a more appropriate synonym. Line 213 needs correction. I suggest “Backed by evidence from Mehboob and colleagues (Mehboob et al., 2011), this study… Line 229 ‘unsolved’ needs correction. Line 235, I suggest authors add ‘Therefore’ after the citation before introducing the statement for the hypothesis. Line 253…other’s… needs correction. Line 256, if CMS is considered as one entity, check ‘affect’ and correct the subject-verb agreement as needed. Authors should correct the sentence in line 259. Introduce linking words between ‘issues’ and ‘meant’. For instance …issues; yet these conflicts are meant… Restructure sentences in lines 261 to 263. Studies…performance. Lines 263 and 266…‘in’… needs correction. Line 277… performance; this …Authors should change this punctuation. Line 285 needs correction. Check and insert the appropriate verb.

2. Literature on the hypotheses (H5a to H5e) for the various mediating variables is not convincing. Your literature refers to counterproductive work behavior. There is no correlation between the literature provided and how the various variables can mediate the relationship between SB and SEP. I missing the point whereby counterproductive can be synonymous with either SB or SEP. The authors need to justify how the dimensions of CMS can mediate the link between SB and SEP. Again, supposedly, H5A to H5e are different from H4. If possible, authors should provide brief literature before they begin to explain these dimensions.

RESEARCH METHODOLOGY

1. Ethical Statement: I suppose your research is only one study, so why do authors use ‘studies’. The whole statement needs to be corrected.

2. Separate the Ethical statement from the methods. Combining them and providing a sub-heading as ‘Ethical Statement’ is confusing.

3. Lines 309 to 314, what are the contributions of these to the research methodology. These sentences should be part of the introduction.

4. The Research Methodology should start from lines 315. Insert the ‘Ethical Statement’ before ‘4. Analysis and Results’.

5. Line 312… (25 percent); overall…Change punctuation. I suggest “(25 percent). Overall… Line 313 is not clear. It needs to be corrected. Line 315 … targets… needs correction. Line 316…SMEs 65.4 percent can be changed to ‘SMEs at 65.4 percent. Line 321 change ‘usable’ to a more appropriate synonym. Lines 322 and 333, The study… concerns… performance, needs correction. Line 330, change …concerning… to make the sentence meaningful.

6. Line 326 to332 should be put under separate sub-heading. For instance, ‘Data Analysis’

7. I missed important details about the data collection procedure in your methodology. Were there ethical considerations regarding your respondents for this study?

8. It is appropriate to insert Table 1 before the Data Analysis. Check the results on the demographic information. The percentage of ‘industry’ is more than 100%.

9. Authors provided a Cronbach alpha value (0.904) under the measure for SEP. However, another Cronbach alpha (0.931) is provided for SEP in TABLE 2. Authors should clarify this. Besides, authors should be consistent in their representation of the results. If they choose to add Cronbach alpha to the measures, as depicted for SEP, then it should be the same for all measures.

ANALYSIS AND RESULTS

1. I suggest authors change the heading to ‘Results’ since there is a suggested sub-heading: ‘Data Analysis’.

2. Line 352…pas… needs correction. Line 380 and 381, insert appropriate punctuation after ‘Scholars’ and revise the subject-verb agreement (Scholars… suggests…) in the sentence. Line 393 needs to be corrected. Line 402, the use of if’ is incorrect. Revise sentence in line 444…show that; H1 shows that…

3. According to Hair’s work, indicator loadings should be equal to or more than 0.70. However, in your paper, you stated a threshold of 0.40 to 0.70, meaning values more than 0.70 is unacceptable. Please refer to the information provided earlier for your major revision, and provide the precise statement.

4. Please provide the full meanings for the abbreviations in Table 2. Do the same for other Tables.

5. Table 3 provides results on the Discriminant validity. Yet Table 5 is entitled Discriminant validity….. It is a bit confusing. Table 5 is on correlations among the constructs, which is different from the correlation to check the discriminant validity. Authors should clarify this. Authors’ denotations of the significant levels are incorrect (eg. ** p-value <0.05).

RESULTS

1. Your direct analysis may be correct. However, authors’ mediation analysis is not acceptable. Please refer to the following works on mediation analysis:

• Preacher, K. J., & Hayes, A. F. (2008). Asymptotic and resampling strategies for assessing and comparing indirect effects in multiple mediator models. Behavior research methods, 40(3), 879-891.

• Zhao, X., Lynch Jr, J. G., & Chen, Q. (2010). Reconsidering Baron and Kenny: Myths and truths about mediation analysis. Journal of consumer research, 37(2), 197-206.

• Shrout, P. E., & Bolger, N. (2002). Mediation in experimental and nonexperimental studies: new procedures and recommendations. Psychological methods, 7(4), 422.

• Nitzl, C., Roldan, J. L., & Cepeda, G. (2016). Mediation analysis in partial least squares path modeling. Industrial management & data systems.

Authors can also refer to David Mackinnon’s works for mediation analysis.

2. Authors provided results on kurtosis and skewness in Table 6. The skewness and kurtosis with their standard errors should be (+ or – 1.96) which indicates normality in the data. But some of the values are more than the threshold. Authors should check and make necessary corrections.

DISCUSSION AND CONCLUSION

1. Line 466…‘supports’… needs correction. Line 467…‘shows’… needs correction. Line 471 ‘effects’… needs correction. Line 489… ‘the five’… needs correction. Line 490…‘individual’… needs correction. The sentence in line 496 needs to be revised. Line 499 …‘affecting’… needs to be corrected. Line 500, Finally, all five…situations, needs to be corrected.

2. Lines 477 to 479 reveal that adaptive performance was affected by SB and SEP. Yet, authors used adaptive performance as a dimension of SEP. I missed the point where authors measured the effect of SEP on adaptive performance. Please clarify this and revise it as needed.

LIMITATIONS AND FUTURE RESEARCH DIRECTIONS

1. In line 522 ‘The time constraint was there to collect more responses’. This is confusing. How were authors able to collect more responses if they were bounded by time?

2. Lines 523 ‘financial aspects forced’… is not clear. Please authors should clarify.

GENERAL COMMENTS

The English language is fairly good. However, the manuscript needs a high level of English language editing. Please go through the whole manuscript to refine the language and make all corrections where necessary. Authors may need a native English speaker to do this properly. There is an article here. However, it demands a major revision.

Reviewer #2: The authors have adequately addressed all the comments raised by the reviewers on the previous manuscript.

Reviewer #3: really its nice research and perfect way to adjust the methodology and conclusion and looking forward for more updates in the research

7. PLOS authors have the option to publish the peer review history of their article (what does this mean?). If published, this will include your full peer review and any attached files.

Reviewer #1: No

Reviewer #2: No

Reviewer #3: Yes: Haitham Medhat Aboulilah

---

## [Author Response · Author response to Decision Letter 1]

8 Jul 2020

REVIEW REPORT

Strengths

The enthusiasm of the authors towards the topic is seen throughout the manuscript. The Manuscript is also detailed and contributes immensely to literature on managing conflicts among employees in SMEs. The authors succeeded in identifying the relevance of their research. However, I found the writing and organization of the manuscript uneven. There are problems in all the sections, including your mediation analysis. 

Weaknesses

ABSTRACT

1. The Abstract will benefit from English Language editing. You mixed up the tenses in the Abstract. For instance, in lines 32 to 34, the purpose…is to investigate…Additionally, the study has examined…Please improve the language.

2. In line 38 “relationship”, needs correction. Lines 43 and 44 need correction. The study…more longitudinal…constructs. Line 45 needs correction. It should read as “…the study adds to the current…

We have addressed the issues in “Abstract” (Lines 32-34, 36-39, 44-46)

INTRODUCTION

1. The Introduction will benefit from English Language editing. For instance, check the preposition after “add”. Check throughout for this problem.

2. In line 71, change the word “primitively” to a more suitable synonym.

3. In line 87, is the mediation role of CMS on SEP, or on the relationship between SB and SEP?

4. Restructure sentence in line 86.

5. We have improved the English language and the issues in “Introduction” 

6. Authors should provide brief information on PLS-SEM in the introduction. 

We have provided brief information on PLS-SEM in the introduction 

LITERATURE REVIEW

1. The study has more than one hypothesis, so correct the spelling in line 98. Check throughout for this problem.

We have addressed the issue (Line 98, 120)

2. In lines 154 and 155 … brake down, and dysfunctional… need correction.

We have addressed the issue (Line 154 and 155)

3. Line 159, you CANNOT start a sentence with a citation: (De-Dreu et al., 1999). Write the name of the author(s), then add the citation. Check throughout for this problem.

We have addressed the issue in Line 159 and checked throughout the manuscript. 

4. Adding the review on the PLS-SEM to the review on the CMS is not right. PLS-SEM and CMS are different. Authors should have a separate sub-heading for PLS-SEM.

We have created a separated sub-heading for PLS-SEM

5. This: “In the literature review, the brief explanation of structural equation modeling and PLS-SEM methodology is missing, i.e., several sentences should be devoted to the methodologies used in the field” should be included as indicated earlier. Authors only showed the reason for using PLS-SEM. You should provide a review of structural equation modeling as well as PLS-SEM methodology.

We have provided a review of structural equation modeling as well as PLS-SEM methodologyin the literature review.

HYPOTHESES DEVELOPMENT

1. Line 194…concerning to… need correction. The sentence in line 196 needs correction. The sentence in lines 203 to 205 needs correction. Line 209, change “ample” to a more appropriate synonym. Line 213 needs correction. I suggest “Backed by evidence from Mehboob and colleagues (Mehboob et al., 2011), this study… Line 229 ‘unsolved’ needs correction. Line 235, I suggest authors add ‘Therefore’ after the citation before introducing the statement for the hypothesis. Line 253…other’s… needs correction. Line 256, if CMS is considered as one entity, check ‘affect’ and correct the subject-verb agreement as needed. Authors should correct the sentence in line 259. Introduce linking words between ‘issues’ and ‘meant’. For instance …issues; yet these conflicts are meant… Restructure sentences in lines 261 to 263. Studies…performance. Lines 263 and 266…‘in’… needs correction. Line 277… performance; this …Authors should change this punctuation. Line 285 needs correction. Check and insert the appropriate verb.

We have addressed the issues.

2. Literature on the hypotheses (H5a to H5e) for the various mediating variables is not convincing. Your literature refers to counterproductive work behavior. There is no correlation between the literature provided and how the various variables can mediate the relationship between SB and SEP. I missing the point whereby counterproductive can be synonymous with either SB or SEP. The authors need to justify how the dimensions of CMS can mediate the link between SB and SEP. Again, supposedly, H5A to H5e are different from H4. If possible, authors should provide brief literature before they begin to explain these dimensions. 

We have revised the literature on the hypotheses (H5a to H5e).

RESEARCH METHODOLOGY

1. Ethical Statement: I suppose your research is only one study, so why do authors use ‘studies’. The whole statement needs to be corrected.

We have corrected the ethical statement.

2. Separate the Ethical statement from the methods. Combining them and providing a sub-heading as ‘Ethical Statement’ is confusing.

We have added “Ethical Statement” as a sub heading before “4. Analysis and Results”.

3. Lines 309 to 314, what are the contributions of these to the research methodology. These sentences should be part of the introduction.

We have removed the Lines and added it in introduction.

4. The Research Methodology should start from lines 315. Insert the ‘Ethical Statement’ before ‘4. Analysis and Results’. 

We have added “Ethical Statement” as a sub heading before “4. Analysis and Results” 

5. Line 312… (25 percent); overall…Change punctuation. I suggest “(25 percent). Overall… Line 313 is not clear. It needs to be corrected. Line 315 … targets… needs correction. Line 316…SMEs 65.4 percent can be changed to ‘SMEs at 65.4 percent. Line 321 change ‘usable’ to a more appropriate synonym. Lines 322 and 333, The study… concerns… performance, needs correction. Line 330, change …concerning… to make the sentence meaningful. 

We have addressed all the issues.

6. Line 326 to332 should be put under separate sub-heading. For instance, ‘Data Analysis’

We have plave the Line 326 to 332 under a separate sub-heading “3.1 Data Analysis”

7. I missed important details about the data collection procedure in your methodology. Were there ethical considerations regarding your respondents for this study? 

We have added “Ethical Statement” as a sub heading before “4. Analysis and Results”

8. It is appropriate to insert Table 1 before the Data Analysis. Check the results on the demographic information. The percentage of ‘industry’ is more than 100%. 

We have inserted “Table 1” before Data Analysis and corrected “industry” percentage. 

9. Authors provided a Cronbach alpha value (0.904) under the measure for SEP. However, another Cronbach alpha (0.931) is provided for SEP in TABLE 2. Authors should clarify this. Besides, authors should be consistent in their representation of the results. If they choose to add Cronbach alpha to the measures, as depicted for SEP, then it should be the same for all measures. 

We have corrected the typing error of Croanbach alpha value and have also added consistent results in all measure

ANALYSIS AND RESULTS

1. I suggest authors change the heading to ‘Results’ since there is a suggested sub-heading: ‘Data Analysis’.

We have changed the heading “Analysis and Results” to “Results” as suggested. 

2. Line 352…pas… needs correction. Line 380 and 381, insert appropriate punctuation after ‘Scholars’ and revise the subject-verb agreement (Scholars… suggests…) in the sentence. Line 393 needs to be corrected. Line 402, the use of if’ is incorrect. Revise sentence in line 444…show that; H1 shows that… 

We have corrected the issues.

3. According to Hair’s work, indicator loadings should be equal to or more than 0.70. However, in your paper, you stated a threshold of 0.40 to 0.70, meaning values more than 0.70 is unacceptable. Please refer to the information provided earlier for your major revision, and provide the precise statement. 

We have corrected the threshold statement as per Hair and Chong, (2017)’s findings. 

4. Please provide the full meanings for the abbreviations in Table 2. Do the same for other Tables.

We have provided the full meaning of abbreviation under table notes of all tables.

5. Table 3 provides results on the Discriminant validity. Yet Table 5 is entitled Discriminant validity….. It is a bit confusing. Table 5 is on correlations among the constructs, which is different from the correlation to check the discriminant validity. Authors should clarify this. Authors’ denotations of the significant levels are incorrect (eg. ** p-value <0.05).

“Table 3” is HTMT ratio. Henseler, Ringle, Sarstedt (2015) suggests to measure the multicollinearity in data by HTMT ratio, which should not be higher than 0.9.

“Table 5” exhibits Discriminant Validity (Latent Variable Correlation & Square Root of AVE).

We have corrected the denotations of the significance levels 

RESULTS

1. Your direct analysis may be correct. However, authors’ mediation analysis is not acceptable. Please refer to the following works on mediation analysis:

• Preacher, K. J., & Hayes, A. F. (2008). Asymptotic and resampling strategies for assessing and comparing indirect effects in multiple mediator models. Behavior research methods, 40(3), 879-891.

• Zhao, X., Lynch Jr, J. G., & Chen, Q. (2010). Reconsidering Baron and Kenny: Myths and truths about mediation analysis. Journal of consumer research, 37(2), 197-206. 

• Shrout, P. E., & Bolger, N. (2002). Mediation in experimental and nonexperimental studies: new procedures and recommendations. Psychological methods, 7(4), 422.

• Nitzl, C., Roldan, J. L., & Cepeda, G. (2016). Mediation analysis in partial least squares path modeling. Industrial management & data systems. 

Authors can also refer to David Mackinnon’s works for mediation analysis.

We have added the mediation analaysis exaplanation as suggested.

2. Authors provided results on kurtosis and skewness in Table 6. The skewness and kurtosis with their standard errors should be (+ or – 1.96) which indicates normality in the data. But some of the values are more than the threshold. Authors should check and make necessary corrections. 

We have made necessary corrections in kurtosis and skewness.

DISCUSSION AND CONCLUSION

1. Line 466…‘supports’… needs correction. Line 467…‘shows’… needs correction. Line 471 ‘effects’… needs correction. Line 489… ‘the five’… needs correction. Line 490…‘individual’… needs correction. The sentence in line 496 needs to be revised. Line 499 …‘affecting’… needs to be corrected. Line 500, Finally, all five…situations, needs to be corrected.

We have revised the issues.

2. Lines 477 to 479 reveal that adaptive performance was affected by SB and SEP. Yet, authors used adaptive performance as a dimension of SEP. I missed the point where authors measured the effect of SEP on adaptive performance. Please clarify this and revise it as needed.

We have revised the explanation.

LIMITATIONS AND FUTURE RESEARCH DIRECTIONS

1. In line 522 ‘The time constraint was there to collect more responses’. This is confusing. How were authors able to collect more responses if they were bounded by time?

2. Lines 523 ‘financial aspects forced’… is not clear. Please authors should clarify.

We have revised the issues in limitations and future research directions.

---

## [Editor Report · Decision Letter 2]

13 Jul 2020

Impact of Supervisory Behavior on Sustainable Employee Performance: Mediation of Conflict Management Strategies using PLS-SEM

PONE-D-20-01676R2

Dear Authors,

We’re pleased to inform you that your manuscript has been judged scientifically suitable for publication and will be formally accepted for publication once it meets all outstanding technical requirements.

Kind regards,

Dejan Dragan, PhD

Academic Editor

PLOS ONE

Additional Editor Comments (optional):

From the Editor’s point of view, the paper has been substantially improved while doing the corrections. All significant issues have been appropriately corrected, and comments have been adequately followed. Moreover, all the Reviewers’ questions and dilemmas have been satisfactorily explained. Accordingly, the AE believes that the paper might have been considered to be accepted and proceeded in the further publishing process.

Academic Editor DD
---

## [Editor Report · Acceptance letter]

17 Jul 2020

PONE-D-20-01676R2 

Impact of Supervisory Behavior on Sustainable Employee Performance: Mediation of Conflict Management Strategies using PLS-SEM 

Dear Dr. Khan:

I'm pleased to inform you that your manuscript has been deemed suitable for publication in PLOS ONE. Congratulations! Your manuscript is now with our production department. 

Kind regards, 

on behalf of

Dr. Dejan Dragan 

Academic Editor

PLOS ONE